# Esterase mutation is a mechanism of resistance to antimalarial compounds

Eva S. Istvan[1,2], Jeremy P. Mallari[1,2,†], Victoria C. Corey[3], Neekesh V. Dharia[3,†], Garland R. Marshall[4], Elizabeth A. Winzeler[3] & Daniel E. Goldberg[1,2]

Pepstatin is a potent peptidyl inhibitor of various malarial aspartic proteases, and also has parasiticidal activity. Activity of pepstatin against cultured *Plasmodium falciparum* is highly variable depending on the commercial source. Here we identify a minor contaminant (pepstatin butyl ester) as the active anti-parasitic principle. We synthesize a series of derivatives and characterize an analogue (pepstatin hexyl ester) with low nanomolar activity. By selecting resistant parasite mutants, we find that a parasite esterase, PfPARE (*P. falciparum* Prodrug Activation and Resistance Esterase) is required for activation of esterified pepstatin. Parasites with esterase mutations are resistant to pepstatin esters and to an open source antimalarial compound, MMV011438. Recombinant PfPARE hydrolyses pepstatin esters and de-esterifies MMV011438. We conclude that (1) pepstatin is a potent but poorly bioavailable antimalarial; (2) PfPARE is a functional esterase that is capable of activating prodrugs; (3) Mutations in PfPARE constitute a mechanism of antimalarial resistance.

[1] Department of Medicine, Division of Infectious Diseases, Washington University School of Medicine, Saint Louis, Missouri 63110, USA. [2] Department of Molecular Microbiology, Washington University School of Medicine, Saint Louis, Missouri 63110, USA. [3] Department of Pediatrics, School of Medicine, University of California San Diego School of Medicine, La Jolla, California 92093, USA. [4] Department of Biochemistry and Molecular Biophysics, Washington University School of Medicine, Saint Louis, Missouri 63110, USA. † Present addresses: Department of Chemistry and Biochemistry, California State University, San Bernardino, California 92407, USA (J.P.M.); Department of Pediatric Oncology, Dana-Farber Cancer Institute and Boston Children's Hospital, Boston, Massachusetts 02215, USA (N.V.D.). Correspondence and requests for materials should be addressed to D.E.G. (email: dgoldberg@wustl.edu).

Resistance to antimalarial drugs is hampering efforts in disease management. The recent emergence of tolerance/ resistance to the artemesinins has given rise to the spectre of failure of artemesinin combination therapies, the mainstay of control regimens[1]. Indeed, resistance has developed to virtually all drugs used for malaria. Resistance mechanisms include efflux pump gene mutation and copy number variation (Mdr1, Crt, Mrp1), enzyme active site mutation (DHFR, DHPS, SoxP, CytC) and Na+/H+ exchanger gene mutation (ATP4) (refs 2,3). There is a pressing need for new antimalarials and for a better understanding of extant and potential resistance mechanisms. Pepstatin is a natural product isolated nearly 50 years ago from *Streptomyces argenteolus* and related actinomycetes[4]. It is a well-characterized aspartic protease inhibitor and derivatives of this scaffold are in clinical use as antiretroviral therapies. Pepstatin has been shown to kill malaria parasites in culture and to cure infection in a rodent malaria model[5–7]. However, the activity of pepstatin against cultured *Plasmodium falciparum* is highly variable depending on the commercial source[8,9].

In this study, we fractionated an active commercial preparation of microbial pepstatin and identified a small contaminant as the active principle, pepstatin butyl ester. By selecting parasite mutants resistant to pepstatin esters, we found that a parasite alpha/beta hydrolase, PfPARE (*P. falciparum* Prodrug Activation and Resistance Esterase), is required for activation of the esterified compound. Esterase mutation is a new mechanism of antimalarial resistance.

## Results

**Isolation and characterization of potent pepstatin analogues.** We used LC/MS to fractionate and characterize the components of a potent pepstatin batch (Fig. 1a). Fractions were tested for antimalarial activity. The dominant peak eluted at 4.5 min and had a mass of 686.47, corresponding to pepstatin. Pure pepstatin was inactive against cultured *P. falciparum* (Fig. 1e). One minor component ($\sim 5\%$ of the total chromatographic signal) that eluted later in the gradient was highly active. Mass spectrometric analysis indicated an m/z of 742.53 for the molecule in the active fraction. Tandem mass spectrometry showed a peptide-like fragmentation that closely resembled pepstatin; however, an extra mass of 56.06, corresponding to a butyl group, was present on the C-terminal statine residue (Fig. 1b–d). We synthesized pepstatin n-butyl ester (PBE) and determined that it is three orders of magnitude more potent than unmodified pepstatin (Fig. 1e).

We tested a series of PBE analogues (Table 1) to probe the structural basis for antimalarial activity. All pepstatin esters were significantly more active than pepstatin, with pepstatin n-hexyl ester (PHE) being the most potent ($EC_{50} = 25$ nM). Penetratin peptide-derivatized pepstatin, as well as pepstatin n-butyl amide, however, were poorly active on blood-stage parasites. N-terminally acetylated pepstatin was also inactive. Our results indicate that esterification at the C-terminus of pepstatin is critical for compound activity and that increasing the length of the ester alkyl chain increases antimalarial potency (hexyl > butyl $\gg$ ethyl $\gg$ methyl).

**Selection of pepstatin ester-resistant parasites.** Selection of mutants resistant to compounds is a powerful way to determine antimalarial mode of action[10,11]. We raised pepstatin ester-resistant parasites by treating $4 \times 10^7$ 3D7 parasites with PBE in a single-step selection. Compound treatment initially cleared cultures of visible parasites and surviving parasites multiplied to detectable levels by 2–3 weeks. Multiple selections yielded viable parasites. Resistant parasites were recloned and tested for sensitivity to pepstatin ester. Selections from different

concentrations of PBE were all similarly highly resistant to pepstatin ester. Four clones were analysed by full-genome sequencing (Table 2). Each had a non-synonymous coding mutation in the same gene, PF3D7_0709700 (*pfpare*). Sequencing of the putative polymorphism-containing gene in other PBE-resistant clones selected from a different parasite strain, HB3, revealed four additional mutations (Supplementary Table 1). Seven of the eight mutations we isolated are missense and one results in an early stop.

**Confirmation of the role of PfPARE mutations in resistance.** We recombinantly modified the parasites in a clean genetic background to confirm that the mutations isolated in resistant parasites provided resistance to pepstatin esters. One of the C-terminal mutations (L357P) was introduced into parasites by single-crossover allelic exchange. Concurrently, we generated recombinants preserving the wild-type sequence. Introduced wild-type or mutant protein is expressed from the endogenous promoter and contains a C-terminal GFP, while the original genomic version is promoterless. Correct integration in clonal parasite lines was confirmed by sequencing of PCR products from integrated plasmids and by Southern blot (Supplementary Fig. 1). GFP-expressing, wild-type parasites were similarly sensitive to PHE compared to the un-transfected, parental parasites, while L357P-expressing parasites were highly resistant to the compound (Fig. 2a).

Live microscopy showed diffuse cytoplasmic GFP fluorescence (Fig. 2b). The intensity of the GFP fluorescence was weak in ring stages and increased with parasite maturity. Western blots of parasite extracts confirmed expression of intact GFP fusion proteins (Fig. 2c). Compared to the wild-type GFP-tagged parasites, the GFP band in the L357P PfPARE parasite extracts was consistently weaker (Fig. 2c).

**PfPARE has sequence similarity to alpha/beta hydrolases.** PfPARE is annotated as a lysophospholipase and is homologous to putative plant lysophospholipases. Structure prediction (Fig. 2d) supports the notion that the esterase belongs to the family of alpha/beta hydrolases, which are enzymes with very diverse specificities. Carboxylic acid esters, epoxides, lipids and phospholipids are substrates of this enzyme class. The gene is present on chromosome seven and not close to the telomeres. However, it belongs to a conserved gene family in *Plasmodium* known as PST-A, many of whose members are subtelomeric[12]. At least three other genes predicted to encode hydrolases closely related to *pfpare* are present in *P. falciparum*: PF3D7_1252600 (54% identical over 349 residues); PF3D7_1401500 (55% identical over 350 residues) and PF3D7_1038900 (53% identical over 338 residues). These three homologous hydrolases contain key catalytic residues and likely encode active enzymes.

Three putative catalytic residues in PfPARE are readily identifiable (S179, D308 and H338). S179 resides in the highly conserved pentapeptide G-x-S-x-G, also known as the 'nucleo-phile elbow'[13]. As in other hydrolases, the core of the alpha/beta fold accommodates less conserved insertions that dictate substrate specificity[14]. Inspection of the structural model in Fig. 2d suggested that some of the resistance-conferring mutations we had obtained likely abolish enzymatic activity. The mutation M180K (clone HB3-R4) resides adjacent to the putative catalytic serine, while in clone 3D7-R4 the predicted protein is truncated and three key catalytic residues (S179, D308 and H338) are missing. Five different mutations cluster in the C-terminal region of the esterase, which is predicted to adopt an alpha-helical conformation. This part of the protein participates in homo-dimer formation in related hydrolases and may stabilize

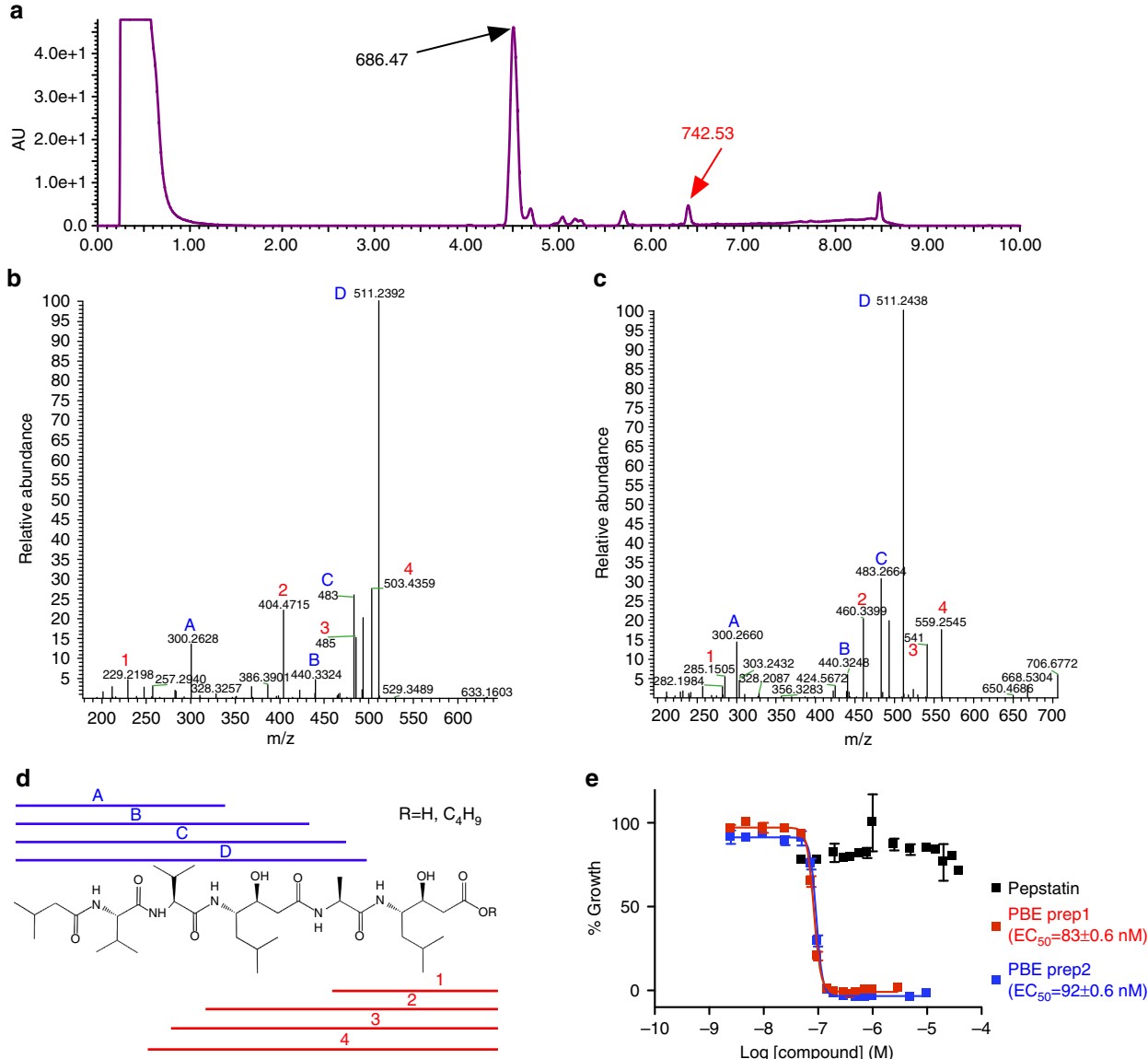

**Figure 1 | Potent pepstatin preparations contain pepstatin esters.** (**a**) Separation of older, potent pepstatin by C18 reverse phase chromatography. The prominent peak eluting at 4.5 min has a mass of 686.47 and does not kill malaria cultures at 40 μM. The minor peak eluting at 6.4 min has a mass of 742.53 and is highly potent. (**b,c**) MS/MS fragmentation of the 686.47 mass peak (**b**) and the 742.53 mass peak (**c**). Mass-to-charge ratios (m/z) of selected fragments are indicated in black numbers. The blue letters and red numbers match fragments of the structure key in (**d**). (**d**) Structure and MS fragmentation key of pepstatin or pepstatin ester. Blue lines and letters mark N-terminal fragments present in the free acid or ester of pepstatin. Red lines and numbers show C-terminal fragments which have an additional 56 amu in the pepstatin ester. (Note that there is loss of a proton on pepstatin upon the addition of the $C_4H_9$ ester.) (**e**) Concentration responses of blood-stage 3D7 parasites (all in 0.1% DMSO) with pepstatin (black) or two syntheses of pepstatin n-butyl ester (PBE) (red, blue) measured using a flow cytometry-based assay. PBE inhibition data were fit to a sigmoidal growth inhibition curve. Error bars show s.d. ± indicates standard errors calculated for the biological triplicates of the experiment shown. $N = 4$.

the enzymes[14]. We observed low GFP fluorescence and decreased signal on western blots from parasite extracts containing the L357P resistance-conferring mutation. Possibly, the enzyme fold is destabilized by a proline-induced break in the C-terminal helix. Alternatively, the change of leucine to proline may slow the folding kinetics of PfPARE, as prolyl cis/trans isomerizations are frequently rate-limiting steps in protein folding[15].

**PfPARE is an active esterase.** To test whether PfPARE activity is important for parasite viability or resistance to pepstatin esters, we generated a parasite clone in which we mutated a predicted key catalytic residue (S179T). The S179T mutation resulted in PHE resistance (Fig. 2a). Active site mutant parasites were also

resistant to other pepstatin esters (Supplementary Table 2). Growth rates were unaffected by either L357P or S179T mutations, indicating that catalytic activity is dispensable for asexual malaria parasites (Supplementary Fig. 2). We assessed whether PfPARE had esterase activity by generating wild-type and active site mutant proteins in *E. coli*. We initially attempted to express the same active site mutant that we had used in the parasite (S179T). However, S179T expressed at low levels, was unstable and could not be purified. We did succeed in expressing another active site mutant (S179G), and this protein was used in subsequent studies. Affinity purified proteins were incubated with the generic esterase substrate p-nitrophenyl butyrate (PNPB), and free p-nitrophenol (PNP) generated by hydrolysis was observed

**Table 1 | Structures and *in vitro* activities of pepstatin and analogues.**

| Compound | Structure | Potency* (μM) |
|---|---|---|
| Pepstatin (P) | Iva-Val-Val-Sta-Ala-Sta-OH | none at 40 |
| Acetyl pepstatin | Ac-Val-Val-Sta-Ala-Sta-OH | none at 20 |
| Pepstatin-Penetratin | Pepstatin-RQIKIWFQNRRMKWKK-OH | none at 1 |
| Pepstatin n-butyl amide | Pepstatin-CO-NH-$(CH_2)_3CH_3$ | none at 20 |
| Pepstatin methyl ester (PME) | Pepstatin-COO-$CH_3$ | 5.12 ± 0.15 |
| Pepstatin ethyl ester (PEE) | Pepstatin-COO-$CH_2$-$CH_3$ | 0.730 ± 0.012 |
| Pepstatin 3-methyl-1-butyl ester (PMBE) | Pepstatin-COO-$(CH_2)_2$CH-$(CH_3)_2$ | 0.071 ± 0.010 |
| Pepstatin n-butyl ester (PBE) prep 1 | Pepstatin-COO-$(CH_2)_3$-$CH_3$ | 0.083 ± 0.006 |
| Pepstatin n-butyl ester (PBE) prep 2 | Pepstatin-COO-$(CH_2)_3$-$CH_3$ | 0.092 ± 0.006 |
| Pepstatin n-hexyl ester (PHE) prep 1 | Pepstatin-COO-$(CH_2)_5$-$CH_3$ | 0.019 ± 0.003 |
| Pepstatin n-hexyl ester (PHE) prep 2 | Pepstatin-COO-$(CH_2)_5$-$CH_3$ | 0.026 ± 0.003 |

$EC_{50}$s of pepstatin and analogues against 3D7 asexual stage parasites measured using a flow cytometry-based assay. Inhibition data were fit to a sigmoidal concentration-response curve using the least squares fit function and without outlier elimination ± are standard errors. $n = 2$–5 (biological replicates performed in triplicate for each experiment).
*Potency—where appropriate, potency is presented as $EC_{50}$. For inactive compounds, inhibition at highest concentration is presented.

**Table 2 | Whole genome sequencing of PBE and MMV011438 resistant parasite clones.**

| Resistant clone | Compound used for selection | Chr | Position | Gene name | Gene description | Effect | Impact | Amino acid |
|---|---|---|---|---|---|---|---|---|
| **3D7-R1** | PBE | 3 | 1066792 | | | Intergenic | | |
| | | 5 | 1290269 | PF3D7_0531600 | 18S ribosomal RNA | Ribosomal RNA | | |
| | | 7 | 435126 | PF3D7_0709700 | Putative esterase | Non synonymous coding | Mis-sense | L357P |
| **3D7-R2** | PBE | 4 | 276762 | PF3D7_0405100 | Sec24 subunit b | Synonymous coding | Silent | S21 |
| | | 4 | 283984 | PF3D7_0405300 | Sequestrin | Synonymous coding | Silent | V1438 |
| | | 7 | 435228 | PF3D7_0709700 | Putative esterase | Non synonymous coding | Mis-sense | L323H |
| **3D7-R3** | PBE | 3 | 1066792 | | | Intergenic | | |
| | | 7 | 435781 | PF3D7_0709700 | Putative esterase | Stop gained | Non-sense | Q139Stop |
| **3D7-R4** | PBE | 6 | 566727 | PF3D7_0613800 | Transcription factor (AP2) | Non synonymous coding | Mis-sense | Q197E |
| | | 7 | 435138 | PF3D7_0709700 | Putative esterase | Frame Shift | Truncation | |
| **Dd2-2G1** | MMV011438 | 5 | 992699 | | | Intergenic | | |
| | | 7 | 435160 | PF3D7_0709700 | Putative esterase | Non-synonymous coding | Mis-sense | N346Y |

Whole genome sequencing from gDNA of compound resistant clones reveals SNPs in PF3D7_0709700. PCR sequencing of PF3D7_0709700 in additional MMV011438 resistant clones indicated that they either had the same mutation as clone 2G1 or a missense (T to G) at bp 435138, resulting in I353S. Filter parameters used to analyse data are in Supplementary Table 3.

photometrically. Wild-type PfPARE was catalytically active, while the S179G mutant protein activity was near background levels (Fig. 2e).

**Esterified pepstatin is taken up by cells and processed.** We performed in-cell metabolism experiments to determine the fate of pepstatin and its ester analogues in parasitized RBCs. Pepstatin or pepstatin esters were incubated with mature ring-stage parasite cultures for 5–6 h. Samples from HPLC-fractionated extracts were analysed by mass spectrometry. Cellular uptake of pepstatin esters and their conversion to free pepstatin could readily be detected in parasite extracts (Fig. 3a). In contrast, cellular accumulation of exogenously added free pepstatin was undetectable. These data suggest that the ionizable carboxylic acid moiety of pepstatin decreases cellular penetration, resulting in poor parasiticidal activity. Replacing the carboxylic acid with an ester results in a neutral molecule, likely resulting in the greatly improved intracellular accumulation and antimalarial potency seen.

After the uptake of pepstatin esters into parental parasites, conversion to free pepstatin was easily detected (Fig. 3b). In contrast, only the pepstatin ester starting material was observed when we used the S179T mutant parasite, indicating that compound processing, but not uptake, was affected by the mutation (Fig. 3c,d). We postulated that PfPARE can act directly on the compounds and tested whether recombinant proteins could hydrolyse pepstatin esters. Affinity-purified wild-type or S179G mutant proteins were incubated with PHE or PME and analysed by LC/MS. Detectable amounts of pepstatin were

generated when wild-type protein was incubated with PHE, but no pepstatin was detected when we instead used the S179G mutant protein (Fig. 3e). Interestingly, only a trace amount of pepstatin was detected when the compound was PME rather than PHE. Previously, we had found PME to be much less potent in parasites than PHE (Table 1). Low potency of short esters could be due to poor penetration or poor activity. The low activity of the recombinant enzyme on PME suggests that esterase activity in the parasite determines efficacy of pepstatin esters.

**PfPARE is active on another antimalarial ester scaffold.** We were curious whether PfPARE could act on ester-containing antimalarials other than pepstatin esters and performed concentration-response experiments with three open source antimalarial compounds. MMV011438, a benzodiazepine, showed differential sensitivity in wild-type and mutant (L357P or S179T) esterase-expressing parasites described above (Fig. 4a), while two other compounds were equipotent (Fig. 4b,c). Incubation of the three compounds with recombinant enzyme revealed hydrolysis of MMV011438 but no modification of the other two compounds (Fig. 3e). MMV030666 and MMV021735 are likely unaffected by activity of PfPARE in the parasite because they are poor substrates. Inactivity on these two compounds is not surprising, as MMV030666 is a tertiary carbamate and MMV021735 is a sterically hindered ester. We raised parasites resistant to MMV011438 as part of the MDTIP consortium[16]. Whole genome sequencing revealed that each evolved clone bore a missense mutation in *pfpare* (Table 2). These data indicate that

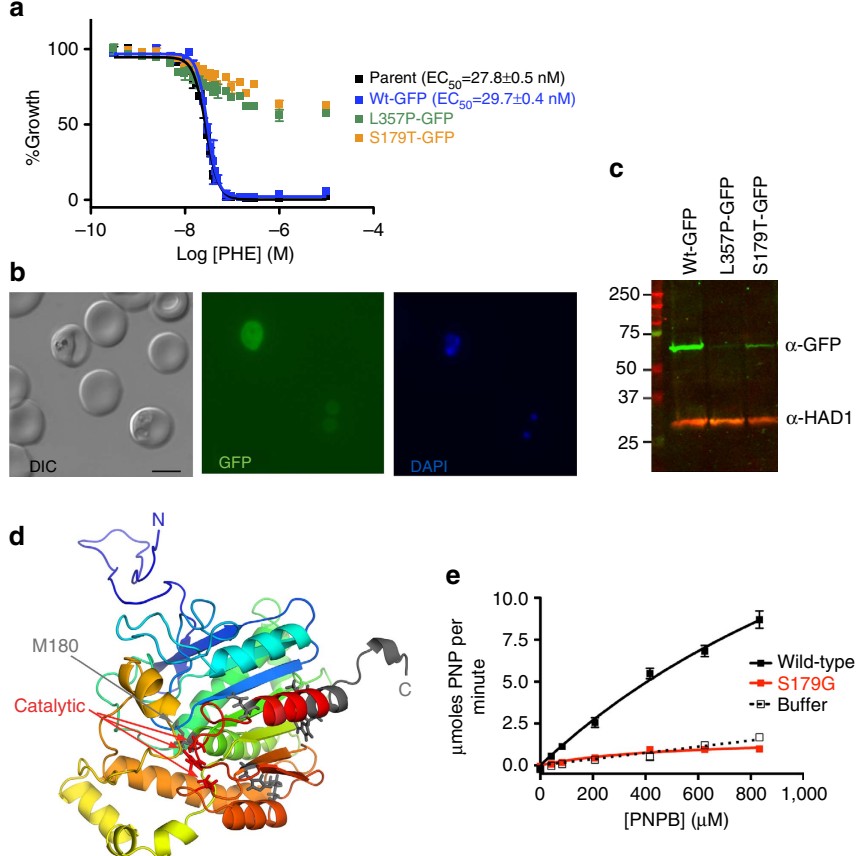

**Figure 2 | Mutations in the esterase PfPARE are responsible for resistance to pepstatin esters.** (**a**) Concentration-responses of parental parasites (black), wild-type allelic replacement (blue), L357P allelic replacement (green) or active-site S179T allelic replacement (orange) to PHE. $EC_{50}$ values were measured with a flow cytometry-based assay. Inhibition data (parent, wild-type) were fit to a sigmoidal concentration-response curve. Error bars show s.d. $\pm$ indicates standard errors calculated for the biological triplicates of the experiment shown. $N = 2$. (**b**) PfPARE is expressed in asexual parasites. Live fluorescence microscopy with GFP-tagged PfPARE shows the protein is in the cytoplasm. Scale bar 5 μm. (**c**) Western blot of wild-type, L357P or S179T GFP-tagged PfPARE using an α-GFP antibody. The protein is expressed from the endogenous promoter. The predicted MW of the PfPARE-GFP fusion is 69 kDa. HAD1 (33 kDa) is a loading control. (**d**) Structural model of PfPARE generated by the Robetta Protein Structure Prediction Server. The gradual colour changes (from blue N-terminus to red) trace the sequence. The side-chains of the three catalytic residues (S179, D308, H338) are drawn as red sticks. Residues mutated in resistant parasites are in grey. (**e**) Recombinant PfPARE has esterase activity on p-nitrophenyl butyrate (PNPB) (black). Purified protein (0.27 μg) was incubated with increasing PNPB and formation of the product p-nitrophenol (PNP) was monitored at 400 nm. Amount of PNP generated as a function of PNPB concentration were fit to a Michaelis-Menten equation. Activity of wild-type protein (black) is significantly different from S179G mutant protein (red) (F-test; $P < 0.05$). Error bars show s.d.

PfPARE is an active esterase with moderate substrate specificity that hydrolyses ester-containing prodrugs.

## Discussion

We have clarified the differences in potency of different pepstatin preparations reported previously and found that antimalarial activity is due not to pepstatin, but to low levels of a highly potent pepstatin ester. It is unclear whether active microbial pepstatin preparations contain pepstatin ester because it is produced by *Actinomycetes* or because pepstatin was modified during compound isolation. We found that esterification is required for parasite uptake of pepstatin, and in this work, we synthesized a series of pepstatin esters, identified structural features important for activity, and developed a potent inhibitor with low nanomolar $EC_{50}$ against cultured *P. falciparum*. Clearly, pepstatin esters are not good lead compounds for drug discovery as resistance arose quickly. Nevertheless, we have gained valuable information: (1) pepstatin likely inhibits the activity of an essential aspartic protease; and (2) the free carboxylic acid moiety is crucial for pepstatin's potency. These findings may help in the development of new antimalarials that do not require esterification for cell uptake.

PfPARE is expressed during the asexual blood stages and resides in the parasite cytoplasm where it may encounter drugs or substrates. Its function is dispensable in blood-stage parasites. The physiological substrate of PfPARE is unknown. Parasites quickly developed resistance to pepstatin esters and to MMV011438 by inactivating the esterase. Given the broad substrate specificity of this enzyme it is likely that PfPARE can hydrolyse a range of other ester-containing antimalarial compounds and that mutations in PfPARE can provide resistance to new potential drugs. It is possible that PfPARE is not essential because other hydrolases are capable of compensating for its physiological function when the esterase is mutated. It is very interesting that these other esterases cannot activate pepstatin esters in the absence of PfPARE. A detailed comparison of substrate specificities would be worthwhile.

To the known antimalarial resistance mechanisms a new one can be added: loss-of-function mutation of a prodrug convertase. There are now thousands of antimalarial hits from whole parasite compound screens. A survey of 125 compounds with antimalarial activity from the pathogen box and 400 compounds from the malaria box reveals 26 esters[17]. Consideration should be given to

the possibility that some are prodrugs subject to rapid resistance development. Testing against parasites with an inactivated esterase gene provides a facile screen to identify such compounds. A limitation of our study is that other non-essential esterases may be active on different compounds. Furthermore, we did not gain full understanding of the substrate specificity of PfPARE.

Resistance to prodrug activation has precedence in the antimicrobial literature. Acyclovir is an anti-herpesvirus nucleoside analogue that is activated by a viral kinase. Since this thymidine kinase is not essential to the virus, inactivation or deletion leads to acyclovir resistance[18]. Isoniazid is a prodrug for tuberculosis that must be activated by the bacterial KatG catalase. Mutations in KatG prevent activation[19]. Metronidazole is a drug that needs to be reduced by the bacterial NADPH-dependent enzyme RDXA for activity against *Helicobacter*[20]. Microaerophilic protists such as *Entameba histolytica* have similar genes, but resistance to metronidazole is achieved by different mechanisms[21]. Esterase inactivation, then, is the first example of a parasitic mechanism to avoid killing by failure to activate a prodrug. Prodrug synthesis is an attractive way to move compounds into cells, but in *Plasmodium*, resistance is potentially a major liability.

## Methods

**Fractionation of potent pepstatin.** Fractionation of microbial pepstatin (Sigma Aldrich, a gift from Hagai Ginsburg, Jerusalem, Israel) was performed on a reverse-phase C18 column using a 0-50% gradient of acetonitrile/$H_2O$ with 0.1% formic acid. Fractions were analysed with MS/MS on a hybrid FTMS consisting of a Thermo LTQ-FT and a Thermo Orbitrap.

**Synthesis of pepstatin analogues.** Acetyl pepstatin, pepstatin methyl ester and pepstatin-penetratin were purchased from Calbiochem. Ethanol, 1-butanol, 3-methyl-1-butanol and 1-hexanol were purchased from Sigma-Aldrich. Esters were synthesized according to the esterification method of Wilcox[22]. Briefly, pepstatin (synthetic, Calbiochem) (10 mg, 15 µmol) was dissolved in the appropriate alcohol (8 g, 80 mmol) and 1 N HCl was added to a concentration of 0.1 N (1 mmol). Reactions were stirred at room temperature for 7–10 days and evaporated under argon. Crude products were dissolved in dimethylsulphoxide and purified by C18 chromatography. Inhibitor concentrations were determined by preparing a standard curve of absorption (218 nm) for pepstatin and calculating the areas under the curve for the pepstatin derivatives. Structures were confirmed by mass spectrometry (measured masses of Na-adducts of PEE = 738.3; PBE = 763.4; PMBE = 780.0; PHE = 793.6). MALDI-TOF spectra of PEE, PBE, PMBE, PHE and pepstatin n-butyl amide are in Supplementary Fig. 3. $^1$HNMR and $^{13}$CNMR peak assignments for PHE are in the Supplementary Note 1. Pepstatin n-butyl amide was synthesized by dissolving 13.5 µmol pepstatin, 27 µmol HBTU and 27 µmol HOBt in a solution of dichloromethane (1 ml) and dimethylformamide (0.5 ml). Three equivalents of n-butylamine was added in DIEA and the reaction was stirred at room temperature for 18 h. The product was purified on HPLC and characterized by MALDI-TOF (mass of 765.2, corresponding to the Na-adduct of the amide) (Supplementary Fig. 3).

**Parasite culture.** 3D7, mycoplasma free, was a gift from P.K. Rathod (University of Washington). The identity of the strain was confirmed by whole-genome sequencing. HB3 was obtained from the MR4 repository at ATCC (Cat. No. MRA-155). Parasites were cultured in human red blood cells (de-identified donors from the St Louis Children's Hospital blood bank) at 2% hematocrit in RPMI 1640 with 0.25% (w/v) Albumax[23,24]. Resistance selections were performed with $4 \times 10^7$ 3D7 or HB3 parasites in a single-step with either 280 nM or 1.2 µM PBE. Resistant parasites were cloned by limiting dilution and two clones for each mutant were analysed. Concentration-response experiments were done in biological triplicates with synchronous, young ring-stage cultures (1–1.2% starting parasitemia) and at least two times. Parasitemias (percentage of total erythrocytes infected with parasites) were measured approximately 70–80 h post compound addition by nucleic acid staining of iRBCs with 0.8 µg ml$^{-1}$ acridine orange in PBS. Growth

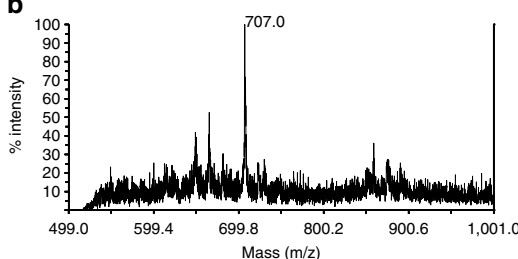

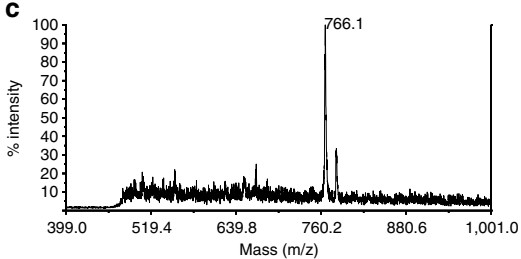

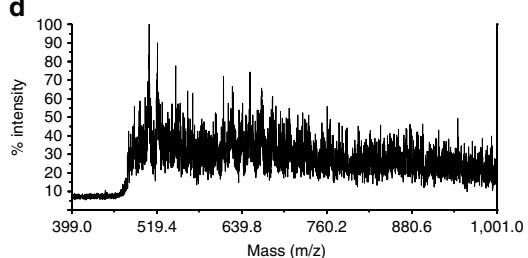

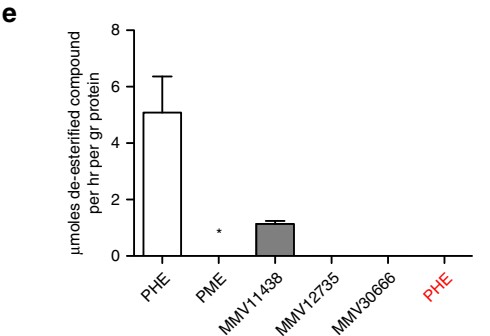

**Figure 3 | In cell metabolism of pepstatin esters. (a)** Detection of pepstatin (P) or pepstatin esters (E) by MALDI-TOF. Fractions of 3D7 lysates after incubation with pepstatin (P) or analogues were spotted onto a MALDI plate and scored for the presence or absence of compounds with appropriate mass. This MS assay is not quantitative. **(b–d)** MALDI-TOF spectra of HPLC fractions (C18) from whole cells extracts of 3D7 **(b)** or S179T mutant parasite **(c,d)** infected RBCs, incubated with PBE. The dominant peak of 707.0 **(b)** corresponds to the Na-adduct of pepstatin. The dominant peak of 766.1 **(c)** corresponds to the Na adduct of PBE. Pepstatin was never detected in any fractions prepared from S179T mutant parasite extracts **(d)**. **(e)** Activity of recombinant PfPARE on pepstatin esters and MDTIP compounds[16]. Wild-type or mutant proteins were incubated with compounds and amounts of de-esterified products were determined by LC/MS (triplicate reactions). Results for incubations with wild-type enzyme are in black, with S179G mutant enzyme is in red. *Pepstatin from incubations with PM could be detected by MALDI-TOF, but not significantly by LC/MS. Error bars are s.d.

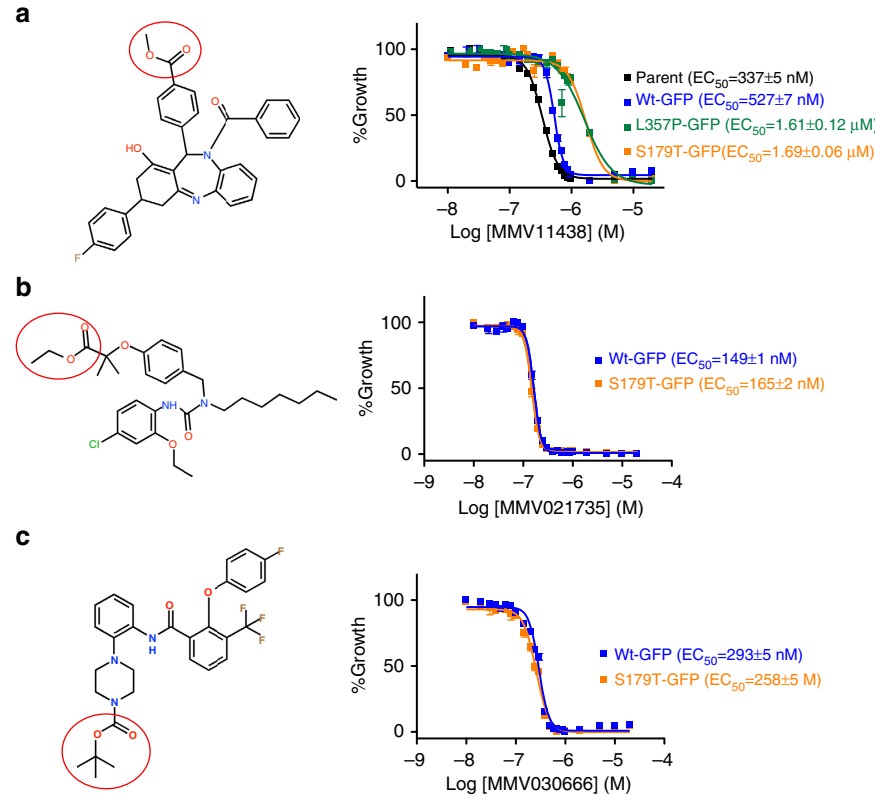

**Figure 4 | PfPARE hydrolyzes and activates an ester-containing prodrug.** Concentration-responses against 3D7 asexual parasites (right) and structures (left) of compounds from the MDTIP consortium[16]. $EC_{50}$ values were measured with a flow cytometry-based assay. Parasite strains are labelled black (3D7), blue (*pfpare* wild-type allelic exchange), green (*pfpare* L357P allelic exchange) and orange (*pfpare* S179T allelic exchange). Inhibition data were fit to a sigmoidal concentration-response curve. (**a**) MMV011438, (**b**) MMV021735, (**c**) MMV030666. In **a**, F-test; $P < 0.0001$ for S179T or L357P compared to Wt. Error bars show s.d. ± indicates standard errors calculated for the biological triplicates of the experiment shown. $N = 2$.

was normalized to parasite cultures with carrier only (dimethylsulphoxide) and chloroquine was used as a 0% growth control. For compounds and strains where inhibition was observed, data were fit to a sigmoidal growth inhibition curve. No curve fitting was performed for resistant parasite clones or inactive compounds. Growth curves for wild-type and mutant *pfpare* parasites were done twice with asynchronous parasite cultures in biological triplicate by measuring daily parasitemias. Data were fit to an exponential growth equation. GraphPad Prism 5.0 was used for data analysis using the least square fit function without outlier elimination and without constraints. Where appropriate, P values (extra sum-of-squares F test) were calculated to evaluate significant differences between samples. Error bars indicate s.d. Concentration-responses of Wt and S179T expressing parasite clones with well-characterized antimalarials are reported in Supplementary Fig. 4. Stage specificity of pepstatin esters was investigated with a PBE washout experiment (Supplementary Fig. 5).

**Genetic modification and analysis.** Allelic exchange constructs were based on the pPM2GT vector[24]. The ORF of PF3D7_0709700 was cloned into the AvrII/XhoI sites using primers (forward = 5'-CTCGAGATGAAGAGCCAGGGTGGAGGG-3' and reverse = 5'-CCTAGGTACTTGTTCTTCTTGTTTGGG-3'). Using this strategy *pfpare* is expressed from the endogenous promoter in-frame with a C-terminal GFP (the native stop is deleted). The mutant constructs were prepared using QuikChange Mutagenesis. The L357P mutagenesis used the primer (5'-GGA ACCTGGAAATGAAAGAGTT**T**TAAAAAAAATTATCACATGGCC-3'; bold letters indicate bp change). This mutant construct contains a synonymous substitution that removes a NheI restriction site at bp 1044. The S179T mutagenesis used the primer (5'-CCATTTTATATTATGGGATTA**A**C**C**ATGGG **A**GGTAATGTTGTTTTAAGAACTTTAC-3'). This mutant construct introduces an NcoI site at bp 537. Transfections were performed in duplicate using 100 µg of supercoiled DNA with 3D7 ring-stage parasites. WR99210 (5 nM) was added to mature ring-stage parasites one cycle after transfection and used to select resistant parasites. Transfectants were cloned by limited dilution. To confirm correct integration, we used the AlkPhos Direct Kit for Southern blots as described (Fisher Scientific)[24]. PCR products for the Southern blots were generated with the following primer pairs (Supplementary Fig. S1A: 5'-GTATAAAATCATATACTA TATCATC-3' and 5'-CATATATTTCCATATTCCCATTTTC-3'; Supplementary Fig. S1C: 5'-CTATCACTTCTAACTATATATGTAAC-3' and 5'-CCATTAGCTA AATTCTTAGGAAGC-3').

**Fluorescence microscopy and western blotting.** Fluorescence microscopy was performed on live, GFP-expressing parasites using a Zeiss Axioskope. Nucleic acid was detected by staining with DAPI. For western blotting, $3 \times 10^8$ parasites were released from iRBCs with 0.035% saponin and lysed in Ripa buffer. For each sample, $9 \times 10^7$ parasites were loaded onto a 12% TGX gel (BioRad). Proteins were transferred onto PVDF using wet transfer with 20% methanol. Blots were blocked overnight at 4 °C with Licor Odyssey block buffer. Primary antibodies were LivingColors mouse-α-GFP (Takara, Cat. No. 632380) (1:500) and rabbit-α-HAD1 (1:1,000) (a gift from Dr Audrey R. Odom, Washington University[25]). Secondary antibodies were goat-α-mouse (800) and donkey-α-rabbit (680) IR-Dyes (1:20,000) from Licor.

**Expression of recombinant PfPARE.** A codon-optimized version (most abundant codon for *E. coli*) of the PF3D7_0709700 gene was purchased from IDT (Supplementary Note 2). The gene product lacked an amino terminal low complexity region that is predicted to be disordered (amino acids 1–22). After cloning into the expression vector pET15-b using the XhoI restriction site, the protein contains an N-terminal hexa-His tag. Including the His-tag and amino acids generated from the vector sequence, the MW of the recombinant enzyme is 42,700. The S179T mutation was introduced using QuikChange Mutagenesis (Agilent) with the primer (5'-CTGCCGTTTTACATTATGGGTCTG**GGG**TATG GGTGGCAACGTTGTGCTGC-3'). Proteins were expressed in C41(DE3) cells (Lucigen; Cat. No. 60442) at 37 °C with 100 µM IPTG for 5 h. Proteins were purified on high-density Ni-resin (Goldbio) in $NaPO_4$ buffer (20 mM, pH 7.2, with 150 mM NaCl). We attempted to use the active site mutant S179T that we had studied in parasite clones as a less-active protein control. However, recombinant expression of S179T esterase only yielded inclusion bodies that could not be successfully refolded. Instead, we generated soluble recombinant protein with the S179G mutation. For assays with p-nitrophenol butyrate (PNPB), 0.27 µg of purified protein was incubated with 0 to 833 µM PNPB (prepared as a 50 mM stock in acetonitrile) in PBS for 5 min at 22 °C. Reactions were stopped by the addition of 100 µl acetonitrile and formation of the yellow hydrolysis product p-nitrophenol (PNP) was monitored at 400 nm. For assays analysed by LC-MS, 35 µg of either wild-type or S179G protein was incubated for 3 h at 22 °C with 5 nmol of compound. The reactions were analysed with a Thermo Scientific TSQ Vantage LC/MS using an isocratic gradient of 60% acetonitrile, 40% $H_2O$ with 5 mM ammonium formate.

**In-cell metabolism of pepstatin esters**. A volume of 150–240 ml of mid-ring parasite cultures at 10–20% parasitemia was incubated with 2 μM compound for 5–6 h. In experiments where comparisons between parasite cultures were important (that is, wild-type and mutant esterase parasite clones), culture amounts and % parasitemia were matched. Compounds not taken up by parasites were removed by permeabilizing infected RBCs with 0.035% saponin and parasites pellets were washed three times with cold PBS. After freeze-thawing, pellets were resuspended in cold 20 mM NaPO$_4$, pH 7.2 (350 μl of buffer per ml of infected culture), slowly mixed at 4 °C for 3 h and extracted with 3 volumes of cold acetonitrile. Centrifugation supernatants (15 min, 2,100$g$) of the extractions were loaded onto a reverse-phase C18 column equilibrated in H$_2$O with 0.1% formic acid and eluted with a linear gradient (solvent B = 100% acetonitrile with 0.1% formic acid). Fractions were collected, mixed 1:1 with α-cyano-4-hydrocinnamic acid, spotted onto a sample plate and analysed on a Voyager 4254 MALDI-TOF.

**Full genome sequencing**. For sequencing, DNA libraries for each gDNA sample were prepped with the Nextera XT kit (Cat. No. FC-131-1024, Illumina), using the dual index protocol. Libraries were clustered using the RapidRun mode on the Illumina HiSeq 2500 and 100 base pairs were sequenced on either fragment end for each sample. Paired-end reads were aligned to the *P. falciparum* 3D7 reference genome (PlasmoDB v. 26.0), as described previously[26] with the exception that mutations were called using GATK's HaplotypeCaller instead of UnifiedGenotyper. Mutations, including single nucleotide polymorphisms and insertion/deletions (INDELS), were filtered based on general recommendations from GATK (Supplementary Table 3). Additional filters were applied post HaplotypeCaller, which included removing positions where read coverage was < 5 in the parent and any position where all samples had a heterozygous ratio > 0.2 (reference/total reads).

**Data availability**. Sequence data have been deposited in the National Center for Biotechnology Information (NCBI) Sequence Read Archive database with accession code SRP077277. Coordinates of the PfPARE model generated by the Robetta Structure Prediction Server are available on modelarchive.org with DOI (10.5452/ma-aqqwe). All other relevant data are available from the authors on request.

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

## Acknowledgements

This study made use of the NIH/NIGMS Biomedical Mass Spectrometry Resource at Washington University in St Louis, MO, which is supported by National Institutes of Health\National Institute of General Medical Sciences Grant # 8P41GM103422. Financial support came from NIH R01 AI112508 (Goldberg), NIH R01 AI103058 (Winzeler) and Gates Foundation Grant OPP 1054480 (Winzeler, Goldberg). We are grateful for the help of Dr Fong Fu-Hsu and Dr Henry Rohrs (Washington University, Saint Louis) with the mass spectrometry experiments. We thank Hagai Ginsburg, Hebrew University, for microbial pepstatin and Audrey R. Odom for the HAD1 antibody. Sha Sha Lu developed the pepstatin esterification protocol in our laboratory and Cindy Choy purified the first preparation of PBE.

## Author contributions

E.S.I. and D.E.G. wrote the manuscript. E.S.I and J.P.M. performed experiments. G.R.M. synthesized pepstatin amide. V.C.C., N.V.D., and E.A.W. performed sequencing and analysed sequence data.

## Additional information

**Competing financial interests**: The authors declare no competing financial interests.

**How to cite this article**: Istvan, E. S. et al. Esterase Mutation is a Mechanism of Resistance to Antimalarial Compounds. *Nat. Commun.* **8**, 14240 doi: 10.1038/ncomms14240 (2017).

**Publisher's note**: 

