## [Peer Review File · Nature Communications]

Reviewers' comments:

Reviewer #1 (Remarks to the Author):

"Esterase mutation is a novel mechanism of resistance to antimalarial compounds".
E.S. Istvan et al.

This report starts with an extraordinary insight on the biochemical mechanisms underlying the potency of a putative protease inhibitor: The antimalarial potency of pepstatin is not due to the parent compound but a small impurity which turns out to be the butyl ester of pepstatin. The authors then systematically show that the antimalarial potency of this compound arises from a non-essential esterase that cleaves pepstatin n-butyl ester (PBE) into pepstatin. Mutations in the gene coding for this esterase abolish the antimalarial potency of the compound and allow parasites to grow without significant fitness costs. The esterase mutants also confer resistance to one ester-based antimalarial compound in the Malaria Box (a collection of possible leads shared in the malaria community as starting points for drug development).

The paper is a truly an excellent exercise in rigorous science: The initial insight leading to the active component in commercial preparations of pepstatine used excellent fractionation and bioassays combined with mass spectrometry and resynthesis of candidate compounds. The in vitro selection for resistance (now an established methods in many labs) was used successfully with whole genome sequencing to show that each independent selection led to a mutation in one and only one gene. In additional truly heroic and non-trivial series of experiments, the authors show that purposeful replacement of the native copy of the PF3D7_0709700 esterase gene with the mutant sequence using molecular genetics conferred resistance to the active ester antimalarial. In vitro expression of native recombinant protein and a 'representative' mutated protein confirmed that the the mutations in the esterase can be responsible for resistance to PBE. The authors further use cell based assays to show that in addition to mutations making dysfunctional esterase the amount of esterase in the cell may also be decreased/destabilized.

The manuscript is important because:

- (i) It shows that pepstatin itself, long thought to be an important antimalarial, is actually not a good antimalarial unless its first derivatized to a butyl ester, and that the hydrolysis of the ester in real time in the presence of parasites is important for its function as an antimalarial.
- (ii) It shows that such ester prodrugs are vulnerable as antimalarials because they rely on endogenous parasites to convert prodrugs into drugs, and the enzyme(s) capable of such conversion can mutate and be dispensable.
- (iii) It demonstrates how a combination of tools (analytical chemistry, biochemistry, molecular genetics, cell biology) can be brought to bear on understanding malaria pharmacology.

Small suggestions for improvement:

(i) In the discussions, it is important to recognize that this is a state of the art basic science study. It formally demonstrates what compounds will not be an antimalarial drug: Pepstatin itself or its derivatives that require a dispensable parasite enzyme for activation of a prodrug.(Line 184-190). Such deep understanding is rare in malaria pharmacology. This reviewer can do without lines 195-196.

(ii) Line 65, the authors start by stating that "To determine the TARGET of pepstatin esters, we raised resistant parasites". Perhaps it would be more appropriate to say "To determine the MODE OF ACTION of pepstatin esters,"

(iii) Activity of pepstatin an antimalarials is known for many decades. Please provide references to key original papers on activity of Pepstatin on malaria parasites (eg J Parasitol. 1981 Oct;67(5):623-6; Z Parasitenkd. 1983;69(3):313-7; Mol Biochem Parasitol. 1986 Mar;18(3):389-400).

(iv) When many other approaches have failed, the use of selection for resistance is now a well-known powerful way to dissect modes of action of antimalarials. Please provide references to key original papers (eg Antimicrob Agents Chemother. 1994 Dec;38(12):2871-6; J Biol Chem. 2005 Apr 8;280(14):13554-9. Epub 2005 Jan 20; Genome Biol. 2009 Feb 13;10(2):R21.; PLoS Pathog. 2013;9(5):e1003375)

(v) The writing style may need an editorial review. Perhaps "we", "to our surprise", etc, can be edited out.

Reviewer #2 (Remarks to the Author):

This is a tight, well written manuscript that will be of great interest to those interested in parasite resistance and drug discovery.

The researchers have discovered that antimalarial activity long attributed to the natural product, pepstatin, is actually due to a contaminant, this being a butyl ester, or at least is a butyl ester in this particular preparation and may be some other ester in other preparations (perhaps the authors could make a statement somewhere in the manuscript about this, if my assumption is true, because this might help others with preparations of pepstatin to not just assume it is a butyl ester that is involved).

The researchers convincingly show this and other esters are bioactive through synthesis and testing. An impressively large swathe of molecular biology is brought to bear, proving that a parasite esterase (PfPARE) is responsible for activating the esters, which in fact are prodrugs, and that resistance rapidly arises from the parasite producing poorly active PfPARE mutants. Live microscopy is undertaken, as well as molecular modelling, which is not always informative but in this case convincingly provides explanations about mutations and resistance.

As apparently the first example of esterase inactivation as a resistance mechanism in parasites, this is a discovery of fundamental significance. Further, it raises questions about one of the most popular concepts in drug discovery for improving cellular activity (make an ester of an acid) for the field malaria drugs, if not more widely to other parasites and

microorganisms.

Some specific comments.

Page 5. L357P mutation indeed could plausibly disorder structure because prolines are known to readily undergo cis-trans isomerism about the amide bond. There is perhaps an opportunity for the authors to therefore strengthen this hypothesis with such a comment and reference.

Page 9. It is fascinating that PfPARE can apparently act on a very different compound, with a benzoyl ester. This indeed suggests that PfPARE might be able to hydrolyse a range of common alkyl and aryl esters. It is noted that neither MMV030666 nor MMV021735 are hydrolysed by PfPARE. Actually, it would be near miraculous if they were. The former would generally be expected to be completely inert, being a tertiary carbamate, while the latter is a highly hindered ester and might also be expected to be inert. In a sense, the authors might have been able to present a stronger case for broad substrate acceptance by testing a few more "standard" esters. This raises the question on the top of page 10 whether some sort of quantitation is required to back up the statement "Consideration should be given to the possibility that some are prodrugs subject to rapid resistance development". A quick substructure search or two of e.g. ChEmbl should inform how many "standard" and "robust" esters in compounds in public databases with antimalarial activity.

In summary, I recommend this manuscript for publication subject to consideration of minor queries above.

Reviewer #3 (Remarks to the Author):

This submission illustrates some laboratory detective work that threw up unexpected findings that yielded to careful investigations to explain them. Moving from commercially available and contaminated preparations of microbial pepstatin to establishing that the ester derivatives in these preparations are the most potent as antimalarials, and that they depend on the actions of specific esterase to exert their increased potency, has required the gamut of modern technologies. The experimental approaches are convincing and appropriate for the questions to be answered, with relevance of findings after assays and whole genome sequencing being bolstered by transfection studies, sub cellular localisations, in vitro anti parasite assays and the use of mass spectrometric techniques to demonstrate that what is proposed as a mechanism is relevant also in parasites. There are some suggestions that could be considered to improve this submission:

1. The paper is a p-value free zone for the most part. Whilst results are robust (as indicated by the errors that apply to mean values), appropriate statistical tests with their results ought to be included wherever conclusions are based on interpretation of differences between experiments. It is unlikely (but this is speculation, of course) that conclusions would be any different, but to demonstrate the robustness of conclusions is worth the extra effort.
2. The analysis of EC50 values needs some more explanation (what results were considered acceptable for analysis after curve fitting in Graphpad - how many were discarded - of any - after experiments were run, and the inclusion of any SOPs that were used please).
3. Whilst directed experiments have demonstrated the generalisability of findings by dipping

into MMV's malaria box and finding one compound that is affected by mutation in esterase that has been identified, it would equally be useful to see that EC50 values against the conventional classes of antimalarials are (are not?) unchanged. Chloroquine is used as a kill control (it is unclear exactly what this is meant to mean - why not just a conventional antimalarial) but a table providing results for artemisinins, and other antimalarial classes as well as chloroquine would be very helpful to contextualise these results. It may be already available.

4. There are no comments on the stage specificity of these effects (both in sensitive and resistant parasite lines) and these would be helpful, bearing in mind that the longer term culture work was carried out in asynchronous parasites.

Some might argue that whilst this is nice science, its relevance to antimalarial resistance for conventional antimalarials is not altogether clear. It seems to me that an entirely new mechanism for resistance is displayed in this work (and one that has been anticipated, at least by this reviewer, based on experience with other microbes and agents) and this mechanism should be highlighted.

Sanjeev Krishna

Reviewer #4 (Remarks to the Author):

Summary

Resistance to antimalarial drug therapy is a growing problem affecting the efficacious treatment of this disorder. A number of resistance mechanisms have been identified to date. In this manuscript, the authors describe a loss of function mutation in parasite esterase (PfPARE) that appears to be a new mechanism for malarial resistance to pepstatin esters. The investigators convincingly show that esterified forms of pepstatin are needed to cross cell membranes and have activity against *P. falciparum*. Once inside the cell, PfPARE hydrolyzes the pepstatin ester prodrugs (inactive) to the active compounds. The authors showed through a series of well-designed and controlled experiments that resistance to these pepstatin analogues rapidly develops via loss-of-function mutations in PfPARE. As a result, the esterified prodrugs are not hydrolyzed to the active antimalarial compound, conferring resistance.

The identification of this novel resistance mechanism can be expected to have important implications for the malaria field, especially relevant to development of antimalarial drugs. These findings show that esterified prodrugs are not likely to have long-term success for treating malaria. The results could inform strategies to modify pepstatin with other structures to facilitate cellular entry but that do not require PfPARE for hydrolysis. Also, models for rapidly screening the activity of new compounds could result from this work. Finally, understanding the molecular basis for this resistance mechanism offers the possibility of developing strategies to enhance or induce the activity of this esterase.

Specific Comments

Page 2 lines 40-42: The authors state that "Prodrug resistance is conferred by mutations in an alpha/beta hydrolase that has esterase activity". Is the hydrolase they refer to PfPARE? If so, please state as such. As written, the text could suggest another enzyme is involved. Also, it would help clarify for readers if the authors also indicate here that this is a loss-of-function mutation and that esterase activity is essential for prodrug hydrolysis and resulting active compound formation.

Page 3 lines 61-63: A brief explanation for why the length of the alkyl ester chain is associated with increased antimalarial potency would be helpful.

Page 7 lines 152-154: Are the intracellular uptake mechanisms passive diffusion or are there active transport mechanisms?

Page 10 lines 221-222: A brief discussion of the limitations of the study is needed.

Page 22 Figure 1E: For the PBE prep1 and prep2 dose response curves, such changing colors to improve readability – the orange and the red are difficult to discern in the curves. Although a minor point here and in several other subsequent figures, this is a concentration-response, not a dose-response, curve.

Page 23 Figure 2A: The same concern here for the gray and black "dose-response" curves as noted above.

We thank the reviewers for their highly constructive comments, which have been addressed as detailed below.

Reviewer #1 (Remarks to the Author):

"Esterase mutation is a novel mechanism of resistance to antimalarial compounds".
E.S. Istvan et al.

This report starts with an extraordinary insight on the biochemical mechanisms underlying the potency of a putative protease inhibitor: The antimalarial potency of pepstatin is not due to the parent compound but a small impurity which turns out to be the butyl ester of pepstatin. The authors then systematically show that the antimalarial potency of this compound arises from a non-essential esterase that cleaves pepstatin n-butyl ester (PBE) into pepstatin. Mutations in the gene coding for this esterase abolish the antimalarial potency of the compound and allow parasites to grow without significant fitness costs. The esterase mutants also confer resistance to one ester-based antimalarial compound in the Malaria Box (a collection of possible leads shared in the malaria community as starting points for drug development).

The paper is a truly an excellent exercise in rigorous science: The initial insight leading to the active component in commercial preparations of pepstatine used excellent fractionation and bioassays combined with mass spectrometry and resynthesis of candidate compounds. The in vitro selection for resistance (now an established method in many labs) was used successfully with whole genome sequencing to show that each independent selection led to a mutation in one and only one gene. In additional truly heroic and non-trivial series of experiments, the authors show that purposeful replacement of the native copy of the PF3D7_0709700 esterase gene with the mutant sequence using molecular genetics conferred resistance to the active ester antimalarial. In vitro expression of native recombinant protein and a 'representative' mutated protein confirmed that the the mutations in the esterase can be responsible for resistance to PBE. The authors further use cell based assays to show that in addition to mutations making dysfunctional esterase the amount of esterase in the cell may also be decreased/destabilized.

The manuscript is important because:

(i) It shows that pepstatin itself, long thought to be an important antimalarial, is actually not a good antimalarial unless its first derivatized to a butyl ester, and that the hydrolysis of the ester in real time in the presence of parasites is important for its function as an antimalarial.

(ii) It shows that such ester prodrugs are vulnerable as antimalarials because they rely on endogenous parasites to convert prodrugs into drugs, and the enzyme(s) capable of such conversion can mutate and be dispensable.

(iii) It demonstrates how a combination of tools (analytical chemistry, biochemistry, molecular genetics, cell biology) can be brought to bear on understanding malaria pharmacology.

Small suggestions for improvement:

(ii) Line 65, the authors start by stating that "To determine the TARGET of pepstatin esters, we raised resistant parasites". Perhaps it would be more appropriate to say "To determine the MODE OF ACTION of pepstatin esters,"

>>Response:
Changed (line 87).

(iii) Activity of pepstatin an antimalarials is known for many decades. Please provide references to key original papers on activity of Pepstatin on malaria parasites (eg J Parasitol. 1981 Oct;67(5):623-6; Z Parasitenkd. 1983;69(3):313-7; Mol Biochem Parasitol. 1986 Mar;18(3):389-400).

>>Response:
Introduced two new references (6 and 7).

(iv) When many other approaches have failed, the use of selection for resistance is now a well-known powerful way to dissect modes of action of antimalarials. Please provide references to key original papers (eg Antimicrob Agents Chemother. 1994 Dec;38(12):2871-6; J Biol Chem. 2005 Apr 8;280(14):13554-9. Epub 2005 Jan 20; Genome Biol. 2009 Feb 13;10(2):R21.; PLoS Pathog. 2013;9(5):e1003375)

>>Response:
Introduced the sentence: "Selection of mutants resistant to compounds is a powerful way to determine antimalarial mode of action." (lines 86-87) and two key references (10,11).

(v) The writing style may need an editorial review. Perhaps "we", "to our surprise", etc, can be edited out.

>>Response:
Edits made.

Reviewer #2 (Remarks to the Author):

This is a tight, well written manuscript that will be of great interest to those interested in parasite resistance and drug discovery.

The researchers have discovered that antimalarial activity long attributed to the natural product, pepstatin, is actually due to a contaminant, this being a butyl ester, or at least is a butyl ester in this particular preparation and may be some other ester in other preparations (perhaps the authors could make a statement somewhere in the manuscript about this, if my assumption is true, because this might help others with preparations of pepstatin to not just assume it is a butyl ester that is involved). The researchers convincingly show this and other esters are bioactive through

synthesis and testing. An impressively large swathe of molecular biology is brought to bear, proving that a parasite esterase (PfPARE) is responsible for activating the esters, which in fact are prodrugs, and that resistance rapidly arises from the parasite producing poorly active PfPARE mutants. Live microscopy is undertaken, as well as molecular modelling, which is not always informative but in this case convincingly provides explanations about mutations and resistance.

As apparently the first example of esterase inactivation as a resistance mechanism in parasites, this is a discovery of fundamental significance. Further, it raises questions about one of the most popular concepts in drug discovery for improving cellular activity (make an ester of an acid) for the field malaria drugs, if not more widely to other parasites and microorganisms.

Some specific comments.

Page 5. L357P mutation indeed could plausibly disorder structure because prolines are known to readily undergo cis-trans isomerism about the amide bond. There is perhaps an opportunity for the authors to therefore strengthen this hypothesis with such a comment and reference.

>>Response:

Introduced the sentence: " the change of leucine to proline may slow the folding kinetics of PfPARE, as prolyl cis/trans isomerizations are frequently rate-limiting steps in protein folding." (lines 148-150). Added reference 15.

Page 9. It is fascinating that PfPARE can apparently act on a very different compound, with a benzoyl ester. This indeed suggests that PfPARE might be able to hydrolyse a range of common alkyl and aryl esters. It is noted that neither MMV030666 nor MMV021735 are hydrolysed by PfPARE. Actually, it would be near miraculous if they were. The former would generally be expected to be completely inert, being a tertiary carbamate, while the latter is a highly hindered ester and might also be expected to be inert. In a sense, the authors might have been able to present a stronger case for broad substrate acceptance by testing a few more "standard" esters. This raises the question on the top of page 10 whether some sort of quantitation is required to back up the statement "Consideration should be given to the possibility that some are prodrugs subject to rapid resistance development". A quick substructure search or two of e.g. ChEmbl should inform how many "standard" and "robust" esters in compounds in public databases with antimalarial activity.

>>Response:

Performed search of Pathogen Box and Malaria Box compounds for ester containing compounds (lines 242-244).

In summary, I recommend this manuscript for publication subject to consideration of minor queries above.

Reviewer #3 (Remarks to the Author):

This submission illustrates some laboratory detective work that threw up unexpected findings that yielded to careful investigations to explain them. Moving from commercially available and contaminated preparations of microbial pepstatin to establishing that the ester derivatives in these preparations are the most potent as antimalarials, and that they depend on the actions of specific esterase to exert their increased potency, has required the gamut of modern technologies. The experimental approaches are convincing and appropriate for the questions to be answered, with relevance of findings after assays and whole genome sequencing being bolstered by transfection studies, sub cellular localisations, in vitro anti parasite assays and the use of mass spectrometric techniques to demonstrate that what is proposed as a mechanism is relevant also in parasites. There are some suggestions that could be considered to improve this submission:

1. The paper is a p-value free zone for the most part. Whilst results are robust (as indicated by the errors that apply to mean values), appropriate statistical tests with their results ought to be included wherever conclusions are based on interpretation of differences between experiments. It is unlikely (but this is speculation, of course) that conclusions would be any different, but to demonstrate the robustness of conclusions is worth the extra effort.

>>Response:

Added p-values to Fig. 2e, Fig. 4 a,b,c and Supplemental Fig. 2. Because no curve fitting was performed for inactive compounds or for highly resistant mutants (Fig. 1e and Fig. 2a) p-values could not be calculated for these experiments.

2. The analysis of EC50 values needs some more explanation (what results were considered acceptable for analysis after curve fitting in Graphpad - how many were discarded - of any - after experiments were run, and the inclusion of any SOPs that were used please).

>>Response:

Updated method section:

GraphPad Prism 5.0 was used for data analysis using the least square fit function without outlier elimination and without constraints. Where appropriate, P-values (extra sum-of-squares F test) were calculated to evaluate significant differences between samples. Error bars indicate standard deviation.

3. Whilst directed experiments have demonstrated the generalisability of findings by dipping into MMV's malaria box and finding one compound that is affected by mutation in esterase that has been identified, it would equally be useful to see that EC50 values against the conventional classes of antimalarials are (are not?) unchanged. Chloroquine is used as a kill control (it is unclear exactly what this is meant to mean - why not just a conventional antimalarial) but a table providing results for artemisinins, and other antimalarial classes as well as chloroquine would be very helpful to contextualise these results. It may be already available.

>>Response:

Clarified in methods section that chloroquine was used as a no growth control (line 305). Provided concentration-response experiments for four antimalarial compounds with wild-type and mutant PfPARE (Supplemental Fig. 3).

4. There are no comments on the stage specificity of these effects (both in sensitive and resistant parasite lines) and these would be helpful, bearing in mind that the longer term culture work was carried out in asynchronous parasites.

>>Response:

Performed stage specificity experiment (Supplemental Fig. 4).

Some might argue that whilst this is nice science, its relevance to antimalarial resistance for conventional antimalarials is not altogether clear. It seems to me that an entirely new mechanism for resistance is displayed in this work (and one that has been anticipated, at least by this reviewer, based on experience with other microbes and agents) and this mechanism should be highlighted.

Sanjeev Krishna

Reviewer #4 (Remarks to the Author):

Summary

Resistance to antimalarial drug therapy is a growing problem affecting the efficacious treatment of this disorder. A number of resistance mechanisms have been identified to date. In this manuscript, the authors describe a loss of function mutation in parasite esterase (PfPARE) that appears to be a new mechanism for malarial resistance to pepstatin esters. The investigators convincingly show that esterified forms of pepstatin are needed to cross cell membranes and have activity against *P. falciparum*. Once inside the cell, PfPARE hydrolyzes the pepstatin ester prodrugs (inactive) to the active compounds. The authors showed through a series of well-designed and controlled experiments that resistance to these pepstatin analogues rapidly develops via loss-of-function mutations in PfPARE. As a result, the esterified prodrugs are not hydrolyzed to the active antimalarial compound, conferring resistance.

The identification of this novel resistance mechanism can be expected to have important implications for the malaria field, especially relevant to development of antimalarial drugs. These findings show that esterified prodrugs are not likely to have long-term success for treating malaria. The results could inform strategies to modify pepstatin with other structures to facilitate cellular entry but that do not require PfPARE for hydrolysis. Also, models for rapidly screening the activity of new compounds could result from this work. Finally, understanding the molecular basis for this resistance mechanism offers the possibility of developing strategies to enhance or induce the activity of this esterase.

Specific Comments

Page 2 lines 40-42: The authors state that “Prodrug resistance is conferred by mutations in an alpha/beta hydrolase that has esterase activity”. Is the hydrolase they refer to PfPARE? If so, please state as such. As written, the text could suggest another enzyme is involved. Also, it would help clarify for readers if the authors also indicate here that this is a loss-of-function mutation and that esterase activity is essential for prodrug hydrolysis and resulting active compound formation.

>>Response:

Clarified (lines 53-56): By selecting parasite mutants resistant to pepstatin esters, we found that a parasite alpha/beta hydrolase, PfPARE (*P. falciparum* Prodrug Activation and Resistance Esterase), is required for activation of the esterified prodrug.

Page 3 lines 61-63: A brief explanation for why the length of the alkyl ester chain is associated with increased antimalarial potency would be helpful.

>>Response:

Added a possible explanation for the association of alkyl ester chain length with increased antimalarial potency:

Previously, we had found PME to be much less potent in parasites than PHE (Table 1). Low potency of short esters could be due to poor penetration or poor activity. The low activity of the recombinant enzyme on PME suggests that esterase activity in the parasite determines efficacy of pepstatin esters. (lines 191-195)

Page 7 lines 152-154: Are the intracellular uptake mechanisms passive diffusion or are there active transport mechanisms?

>>Response:

Unknown.

Page 10 lines 221-222: A brief discussion of the limitations of the study is needed.

>>Response:

Added: “A limitation of our study is that other non-essential esterases may be active on different compounds. Furthermore, we did not gain full understanding of the substrate specificity of PfPARE.” (lines 246-248).

Page 22 Figure 1E: For the PBE prep1 and prep2 dose response curves, such changing colors to improve readability – the orange and the red are difficult to discern in the curves. Although a minor point here and in several other subsequent figures, this is a concentration-response, not a dose-response, curve.

>>Response:

Changed colors from orange and red to blue and red in Fig. 1e. Changed dose-response to concentration-response throughout manuscript.

Page 23 Figure 2A: The same concern here for the gray and black “dose-response” curves as noted above.

>>Response:

Changed colors to black and blue.

REVIEWERS' COMMENTS:

Reviewer #2 had no more comments for the authors

Reviewer #3 (Remarks to the Author):

The manuscript has been appropriately revised and has dealt with reviewers' comments, all of which were supportive. The additional data provided in supplements is also useful.

Reviewer #4 (Remarks to the Author):

The authors provided detailed responses and revisions based on the questions/concerns in the original manuscript. I have no additional questions or concerns about the revised paper.